# AutoMoMa:
# Scalable Coordinated Mobile Manipulation Trajectory Generation

## Abstract

Mobile robots need coordinated whole-body motion to perform household tasks effectively. Current mobile manipulation datasets rely on expensive teleoperation or slow planning methods, limiting available data to hundreds of demonstrations. This data scarcity severely constrains the development of generalizable learning-based policies. Here, we demonstrate that GPU-accelerated planning generates up to 5,000 episodes per GPU hour, over $80\times$ faster than existing methods. Our AutoMoMa pipeline produces 500K diverse physically valid whole-body motions across 300 household scenes and multiple robot embodiments, compared to previous datasets limited to narrow robot-scene pairs with a few hundred demonstrations. Downstream validation demonstrates consistent policy improvements with large-scale training data. This work provides the first scalable solution to the mobile manipulation data bottleneck. By enabling massive dataset generation, AutoMoMa accelerates progress toward general-purpose household robots capable of complex coordination tasks.

## 1 Introduction

Mobile manipulation is a fundamental capability for autonomous robots operating in unstructured human environments. Unlike fixed-base manipulators with limited reach, mobile manipulators combine the mobility of a base with the dexterity of an arm, thereby extending the effective workspace and enabling interaction with a diverse range of household objects. Achieving this capability requires **coordinated whole-body motion**, as many real-world tasks inherently couple navigation and manipulation, requiring simultaneous planning over base positioning, arm configuration, and object interaction (Khatib, 1999; Mittal et al., 2022; Sleiman et al., 2023).

Traditional approaches typically decompose mobile manipulation into separate navigation and arm control stages or rely on handcrafted coordination strategies for specific tasks (Sleiman et al., 2023). While effective in constrained setups, these methods require extensive manual effort to encode task constraints and generalize poorly across diverse settings. Recent learning-based methods show promise for end-to-end whole-body policies (Li et al., 2023; Zhang et al., 2024), but their progress is hindered by the lack of large-scale, diverse datasets capturing physically valid coordinated motions.

Although several large-scale manipulation datasets have emerged, mobile manipulation datasets remain highly limited (Tab. 1). Existing efforts often oversimplify the problem by restricting to static tabletop scenes (Geng et al., 2023; Cui et al., 2025), focusing on a single robot embodiment (Pari et al., 2021; Bahl et al., 2023), or targeting narrow task classes (Wu et al., 2023). As a result, they lack the scale, task diversity, and coordinated base-arm-object interactions necessary to train generalizable whole-body policies.

These shortcomings largely arise from their collection methodologies. Reinforcement Learning (RL) requires prohibitively expensive trial-and-error exploration (Fu et al., 2023; Xia et al., 2021; Li et al., 2023), especially when scaling across object variations and environments; teleoperation (Fu et al., 2024) is bottlenecked by expert availability and hardware interface limitations; and traditional planning-based methods require sequential planning of base and arm, which failed to capture coordinated motion. The Augmented Kinematic Representation (AKR) framework (Jiao et al., 2025) offers a principled way to unify base, arm, and object kinematics in a single representation (Jiao

Figure 1: AutoMoMa pipeline for generating large-scale coordinated mobile manipulation trajectories. By combining AKR modeling with GPU-accelerated planning, the system produces diverse, physically valid whole-body motions across robots, tasks, and household scenes. These data support the training of modern learning-based policies, including DP, DP3, and ACT.

et al., 2021a;b), but existing implementations are computationally intensive, resulting in generation speed at around 60 effective trajectories per hour (Zhang et al., 2024), limiting the ability to generate scalable datasets. These constraints have fragmented research efforts, forcing teams to develop narrow-purpose datasets (Pari et al., 2021; Schiavi et al., 2023; Ceola et al., 2023; Wu et al., 2023; Fu et al., 2024) that fail to capture the full spectrum of mobile manipulation scenarios, ultimately impeding progress toward general-purpose household robots.

We address these challenges with AutoMoMa, a scalable framework for generating diverse, high-quality whole-body mobile manipulation trajectories. By combining AKR-based modeling with GPU-accelerated motion planning (Sundaralingam et al., 2023), AutoMoMa generates physically valid trajectories at 5,000 per GPU-hour—orders of magnitude (80×) faster than prior approaches. This efficiency enables large-scale data generation across multiple robots, articulated objects, and realistic household scenes without requiring costly human demonstrations. Beyond simple pick-and-place, AutoMoMa supports articulated-object interactions and multi-step tasks with grasp switching in confined spaces. We validate its effectiveness both in simulation and on a dual-UR5 Clearpath Ridgeback platform, demonstrating successful sim-to-real transfer.

Our contributions are:

- **Scalable pipeline:** A GPU-accelerated AKR planner with 80x faster trajectory generation (up to 5k trajectories per GPU-hour), enabling scalable whole-body data collection.
- **Comprehensive dataset:** 500k trajectories across 300 photorealistic scenes, 26 articulated objects, 3 robot morphologies, and 150 tasks, with straightforward extensibility.
- **Benchmark foundation:** Baselines with learning-based policies (DP, DP3, ACT) and real-robot validation to support learning methods and sim-to-real research.

Together, these contributions establish AutoMoMa as the first scalable framework that bridges high-performance planning with large-scale dataset generation for coordinated mobile manipulation.

## 2 RELATED WORK

### 2.1 MOTION PLANNING FOR MOBILE MANIPULATION

**Model-Based Planning** Classical approaches to coordinated mobile manipulation include task-specific controllers such as impedance and model-predictive control for doors and drawers (Jain and Kemp, 2010; Karayiannidis et al., 2016; Stuede et al., 2019), as well as general base-arm optimization frameworks for cluttered environments (Berenson et al., 2008; Gochev et al., 2012; Bodily et al., 2017). While effective under controlled conditions, these methods require extensive hand-tuning for each robot-object pair and do not scale well to diverse environments or object types. The AKR framework (Jiao et al., 2021a;b; 2025) advanced the field by unifying the base, manipulator, and object into a single kinematic model, enabling constraint-aware planning in a unified configuration

Table 1: **Comparison of mobile manipulation datasets from the Open-X-Embodiment project (**O'Neill et al., 2024**).** Existing datasets are generally limited in scale, scene diversity, or ability to capture coordinated whole-body joint trajectories, and most rely on human demonstrations or scripted policies. "EEF" denotes *end-effector*.

| Dataset | Robot | #Episodes | Coordinated Motion | #Scenes | Action Space | Data Collection Method |
|---|---|---|---|---|---|---|
| RT-1 Robot Action (Brohan et al., 2022) | Google Robot | 73,499 | Yes | 10 | EEF Position | Human VR |
| NYU VINN (Pari et al., 2021) | Hello Stretch | 435 | Yes | 3 | EEF Position | Human Kinesthetic |
| BC-Z (Jang et al., 2022) | Google Robot | 39,350 | Yes | 2–3 | EEF Position | Human VR |
| ETH Agent Affordances (Schiavi et al., 2023) | Franka | 120 | No | 50 | EEF Position | Expert Policy |
| QUT Dexterous Manipulation (Ceola et al., 2023) | Franka | 200 | No | 1 | EEF Position | Human VR |
| CMU Stretch (Bahl et al., 2023; Mendonca et al., 2023) | Hello Stretch | 135 | No | 10 | EEF Position | Expert Policy |
| ConqHose (Mitrano and Berenson, 2024) | Spot | 139 | Yes | 3 | EEF Velocity | Scripted |
| DobbE (Shafiullah et al., 2023) | Hello Stretch | 5,208 | Yes | 216 | EEF Position | Human Tools |
| MobileALOHA (Fu et al., 2024) | MobileALOHA | 276 | Yes | 5 | Joint Position | Human Puppeteering |
| TidyBot (Wu et al., 2023) | TidyBot | 24 | No | 104 | Other | Handcrafted Object Placements |
| **Ours** | **Franka, R1, TIAGo** | **500,000** | **Yes** | **300** | **Joint Position** | **Physics Plausible Motion Planner** |

space. This approach naturally handles articulated objects, but conventional CPU-based implementations remain computationally expensive, assume fixed grasp poses, and are limited in task diversity (Zhang et al., 2024).

**Learning-Based Planning**   End-to-end deep RL has been applied to coordinated base-arm control in simulation (Xia et al., 2021; Fu et al., 2023), but remains highly sample-inefficient and struggles to generalize across novel robots or environments (Sun et al., 2022). Imitation learning provides a more data-efficient alternative by leveraging demonstrations (Fu et al., 2024; Jang et al., 2022), yet it is constrained by dataset scale and diversity. Both approaches are ultimately bottlenecked by the scarcity of large, high-quality datasets capturing realistic whole-body coordination. This motivates the need for scalable data generation platforms like AutoMoMa that can bridge the gap between planning frameworks and data-driven methods.

## 2.2   DATA COLLECTION FOR MOBILE MANIPULATION

**Simulated Embodied AI Platforms**   Platforms such as Habitat 2.0 (Szot et al., 2021), AI2-THOR (Kolve et al., 2017), OmniGibson (Li et al., 2023), and RoboHive (Kumar et al., 2023) provide photorealistic environments with articulated assets, but typically prioritize visual realism over physically valid robot motion. Interactions are often simplified to scripted primitives that bypass base-arm kinematics and whole-body coordination. ManiSkill-HAB (Shukla et al., 2025) makes progress with 8,000 demonstrations of coordinated table-setting, but is limited to a single kitchen and narrow task diversity.

**Teleoperation**   Human-guided teleoperation captures realistic behaviors but scales poorly. Early systems like MOCA (Wu et al., 2019) and MOMA-Force (Yang et al., 2023) recorded only end-effector trajectories, omitting full joint-space motion. Recent platforms such as Mobile ALOHA (Fu et al., 2024) and TeleMoMa (Dass et al., 2024) collect high-fidelity joint-space data, yet remain constrained by operator fatigue, hardware availability, and limited environment diversity, restricting datasets to thousands rather than millions of demonstrations.

**Standalone Mobile Manipulation Datasets**   Despite increasing interest, large-scale datasets for mobile manipulation remain scarce. BC-Z (Jang et al., 2022) includes 25,000 demonstrations but mostly involves stationary bases and end-effector poses only. Mobile ALOHA (Fu et al., 2024) contributes 276 joint-space demonstrations coupling a base with a 7-DoF arm, but is limited to a single platform and lacks scale. Overall, current datasets are narrow in task coverage, robot diversity, and physically valid coordination.

In contrast, our AutoMoMa platform enables scalable, automated generation of diverse, constraint-compliant whole-body trajectories across multiple robots, articulated objects, and realistic environments—providing the breadth and quality of data required to advance learning-based mobile manipulation.

## 3   PRELIMINARY

This section briefs AKR-based mobile manipulation planning, illustrating how the AKR formulation enables scalable whole-body trajectory generation for manipulating both rigid and articulated

objects. We begin by introducing the AKR modeling, then formulate motion planning problems from the AKR perspective, and finally describe how task and environmental constraints are incorporated into this framework.

## 3.1 AKR MODELING

The objective of the modeling is to construct a serial AKR by composing the kinematics of the mobile base, manipulator arm, and the object to be manipulated (Jiao et al., 2025). This requires three inputs: 1) the robot's kinematic tree, 2) the object's kinematic tree, and 3) the transformation between the robot's end-effector and the object's attachable frame (*i.e.*, the grasping pose). The procedure for constructing the AKR consolidates the robot and object kinematic models as follows.

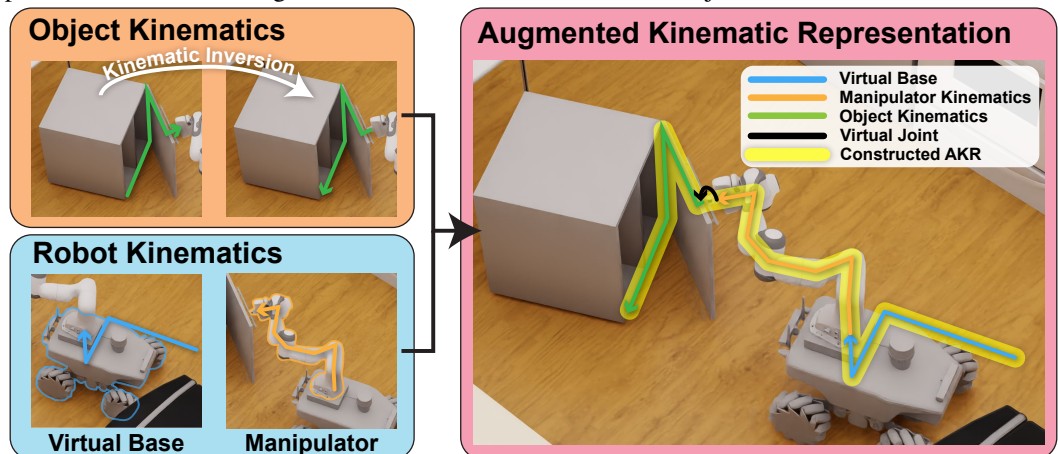

Figure 2: **Augmented Kinematic Representation (AKR) Construction.** The AKR unifies the kinematics of the mobile base, manipulator, and object into a single serial chain by introducing a virtual base and a virtual joint that connects the manipulator to the object. For articulated objects (*e.g.*, cabinets), the object model is inverted to maintain a valid serial structure suitable for trajectory optimization.

The kinematic structures of the robot and the object are each represented as separate kinematic trees (*e.g.*, Unified Robot Description Format (URDF)), as shown in Sec. 3.1. To form a serial AKR, we insert a virtual joint, corresponding to the grasping pose, between the robot's end-effector and the object. This requires inverting the object's kinematic model. Importantly, inverting a kinematic tree is not simply a matter of reversing the parent-child relationships; all associated transformations, including those branching structures, must be carefully updated, as revolute and prismatic joints typically define motion relative to the child link's frame. Moreover, the geometry of branching structures must also be considered during trajectory optimization to ensure safety and feasibility.

To jointly optimize locomotion and manipulation, we further insert a virtual base that models the mobile base's planar motion. This is implemented using two orthogonal prismatic joints and one revolute joint between the virtual base and the robot's base, allowing for planar motion while preserving a serial kinematic structure.

Sec. 3.1 illustrates a constructed AKR for a door-opening task. The resulting AKR begins with a fixed virtual link and ends at the object link connected to the ground (*e.g.*, a door's frame). The mobile base and manipulator are embedded within this serial chain. Consequently, the states of the mobile base, arm, and object are jointly represented within the AKR configuration space. Task goals and kinematic constraints are subsequently imposed during trajectory optimization, as described in the next section.

## 3.2 AKR-BASED MOBILE MANIPULATION PLANNING

The mobile manipulation planning problem can be modeled as finding a collision-free trajectory within the configuration space of the AKR. Formally, the resulting AKR state is defined as:

$$\boldsymbol{x} = \left[\boldsymbol{q}_B^\mathsf{T}, \boldsymbol{q}_M^\mathsf{T}, \boldsymbol{q}_O^\mathsf{T}\right]^\mathsf{T} \in \mathcal{X}_{\text{free}}, \tag{1}$$

where $\boldsymbol{q}_B \in \mathbb{R}^3$ is the mobile base pose, $\boldsymbol{q}_M \in \mathbb{R}^n$ is the manipulator joint state ($n$ is the Degree of Freedom (DoF) of the manipulator), $\boldsymbol{q}_O \in \mathbb{R}^m$ is the articulated object's joint state ($m$ is the DoF of

Figure 3: **AutoMoMa data generation pipeline.** The pipeline begins by preprocessing planning contexts through AKR construction and collision processing. It then models mobile manipulation tasks from the AKR perspective and solves them via trajectory optimization. Finally, the data undergoes postprocessing to enforce constraints and generate multi-modal outputs.

the articulated object, 0 for rigid object), and $\mathcal{X}_{\text{free}}$ is the collision-free configuration space. Then, the motion planning problem seeks a collision-free path of length $T$: $\boldsymbol{x}_{1:T} = \langle \boldsymbol{x}_{[1]}, \dots, \boldsymbol{x}_{[T]} \rangle \subset \mathcal{X}_{\text{free}}$.

During trajectory optimization (following Jiao et al. (2025)), we enforce:

$$h_{\text{chain}}(\boldsymbol{x}_{[t]}) = 0, \qquad\qquad \forall t = 1, \dots, T, \qquad (2)$$

$$\|f_{\text{task}}(\boldsymbol{x}_{[T]}) - \boldsymbol{g}_{\text{goal}}\|_2^2 \leqslant \xi_{\text{goal}}, \qquad (3)$$

$$\boldsymbol{x}_{\min} \leqslant \boldsymbol{x}_{[t]} \leqslant \boldsymbol{x}_{\max}, \qquad\qquad \forall t = 1, \dots, T, \qquad (4)$$

$$\|\Delta \boldsymbol{x}_{[t]}\|_\infty \leqslant \Delta \boldsymbol{x}_{\max}, \qquad\qquad \forall t = 1, \dots, T-1, \qquad (5)$$

$$\|\Delta \dot{\boldsymbol{x}}_{[t]}\|_\infty \leqslant \Delta \dot{\boldsymbol{x}}_{\max}, \qquad\qquad \forall t = 2, \dots, T-1. \qquad (6)$$

Here, Eq. (2) enforces the physical constraints arising from robot-environment interactions (*e.g.*, a door hinged to the ground, or a chair constrained to planar motion along the floor); Eq. (3) bounds the task goal via $f_{\text{task}} : \mathcal{X} \to \mathcal{G}$ with tolerance $\xi_{\text{goal}}$; and Eq. (4)-Eq. (6) impose joint limits, velocity, and acceleration bounds. Collision avoidance is handled by the underlying motion planner's self- and environment-collision checks.

## 4 DATA GENERATION PIPELINE

Our data generation pipeline consists of four stages. First, we prepare the task description by loading the environment, robot embodiment, objects, and start/goal configurations into the planning scene. Second, we prepare the planner input by calculating voxelized environment collision models, constructing AKR representations for robot-object pairs, and create sphere-based collision approximations. Third, we solve the trajectory optimization problem, where the AKR start and goal states are computed and whole-body trajectories are generated under task-specific constraints. Finally, we post-process the optimized trajectories and render multi-modal outputs in Isaac Sim, including RGB-D images and both egocentric and fixed-viewpoint point clouds. Detailed information about our scene generation process is described at Appx. C

### 4.1 TASK DESCRIPTIONS

The AutoMoMa pipeline receives a triplet $(\mathcal{S}, \mathcal{O}, \mathcal{R})$ that jointly defines the motion planning problem: a *scene* $\mathcal{S}$, a finite set of *interactive objects* $\mathcal{O}$, and a *robot embodiment* $\mathcal{R}$, together with start and goal configurations for each task.

**Household scene layouts.** Each scene $\mathcal{S}$ specifies the geometry, appearance, and semantic tags for static elements such as floors, walls, countertops, and fixed appliances. A world frame is anchored at the scene's geometric center, and all elements include both visual and collision meshes for rendering and collision checking. To enrich environmental diversity, scenes are constructed either by procedurally generating new layouts with articulated objects or by augmenting existing layouts through the replacement of static assets with articulated counterparts.

**Interactive objects:** The object set $\mathcal{O} = \mathcal{O}_{\text{rigid}} \cup \mathcal{O}_{\text{art}}$ contains *rigid bodies* $O_{\text{rigid}}$ and *articulated objects* $O_{\text{art}}$. Every rigid object $o \in \mathcal{O}_{\text{rigid}}$ consists of a watertight mesh and a set of grasp poses expressed in the object's base frame. For each articulated object $o \in \mathcal{O}_{\text{art}}$, we require a URDF specifying joint types, axes, limits, and inertial parameters. Grasp poses are state-dependent—for example, a closed cabinet may afford different grasps than when open. The articulated objects are inverted for AKR modeling, as introduced in Sec. 3.1.

**Robot embodiments:** A robot embodiment $\mathcal{R}$ consists of a virtual mobile base and a manipulator. Both are defined following URDF; an auxiliary file provides (i) a spherical approximation of collision geometry, (ii) a self-collision mask, and (iii) a joint-weight vector $\boldsymbol{w} \in \mathbb{R}^{n+m+3}$ used by the trajectory optimization. Any robot embodiment that satisfies the above description can be loaded without further modification. This paper has validated it on a Franka arm mounted on a Summit base, the R1 robot adopted from OmniGibson (Li et al., 2023), and the Tiago model from PAL Robotics.

## 4.2 AKR Construction and Collision Processing

Manipulated objects are integrated into robotic manipulation pipelines via the following workflow.

**Preprocessing:** Since standalone datasets typically provide objects at a fixed scale, we resize them to fit the scene and update grasp poses accordingly. In this process, the geometric components of each link are merged into a single mesh and scaled accordingly. Since scaling alters the spatial configuration, joint origins are updated to preserve valid kinematic relationships.

**AKR construction:** To construct a AKR, the post-inversion object model is treated as an extended limb of the robot, The grasp pose defines the transformation between the robot's end-effector and the target object link. This transformation, along with the two associated links, forms a virtual joint that connects the object to the robot, yielding a unified kinematic model $\mathcal{K}_{\text{akr}}$ for integrated motion planning.

**Collision Processing:** To enable efficient collision checking in the GPU-accelerated motion planner, each AKR's geometry is approximated using a set of fitted spheres. To avoid overestimating the original shape, the merged mesh is slightly downscaled before fitting spheres to its geometry. In rare cases of translational shifts, the sphere cloud's centroid is aligned with that of the original mesh to preserve geometric consistency. Finally, we identify negligible collision pairs (*e.g.*, adjacent links that are always in contact) in the AKR, ensuring efficient collision checking.

**Environment Collision Models:** Each scene is converted into an Euclidean Signed-distance Field (ESDF) to accelerate collision checking. During planning, only the ESDF voxels within an axis-aligned bounding box, defined by the target object's start and goal states, are considered, further limiting collision checks to the local workspace and reducing unnecessary computations.

## 4.3 Trajectory Generation

We generate trajectories by solving an optimization problem in the AKR space, which jointly optimizes the base, arm, and object states. This enables object-centric goals and task-specific constraints, and supports both rigid-object relocation and articulated-object manipulation with grasp switching when needed.

**Defining Task Objectives and Goals:** The mobile manipulation planning objective minimizes total traveling distance and trajectory non-smoothness:

$$\mathcal{J}(\boldsymbol{x}_{1:T}) = \sum_{t=1}^{T-1} \left\| \boldsymbol{w}_v \, \Delta \boldsymbol{x}_{[t]} \right\|_2^2 + \sum_{t=2}^{T-1} \left\| \boldsymbol{w}_a \, \Delta \dot{\boldsymbol{x}}_{[t]} \right\|_2^2, \tag{7}$$

$$\boldsymbol{x}_{1:T}^{\star} = \arg\min_{\boldsymbol{x}_{1:T}} \mathcal{J}(\boldsymbol{x}_{1:T}). \tag{8}$$

where $\boldsymbol{w}_v$ and $\boldsymbol{w}_a$ are diagonal weight matrices over each DoF, enabling modulation of base-arm coordination strategies. Task goals are object-centric: for rigid-object relocation, the goal is the grasp pose to the object or a target placement pose of the object; for articulated objects, the goal is a desired object state (*e.g.*, a door opened to a specific joint angle).

**Specifying Task Constraints:** Trajectory constraints are defined based on the object-scene relationship and task type. For rigid object relocation, the object is treated as a free joint. For tasks involving large objects or specific task requirements—such as pushing a chair or sweeping a table—we impose planar constraints on the AKR's end-effector (*i.e.*, the object's base link) to ensure stable, planar motion. When manipulating articulated objects fixed to the environment, we enforce a fixed constraint on the AKR's end-effector, penalizing deviations of the object's location from its initial pose via a pose cost.

**Start and Goal Configurations:** To initialize the motion planning problem, we compute start and goal AKR configurations under the assumption of a fixed grasp pose during execution. These configurations are obtained by solving Inverse Kinematics (IK) for both object states. Similar AKR configurations are removed through clustering, yielding a compact yet diverse set of candidate configurations. This reduces planning overhead while maintaining broad workspace coverage, facilitating efficient trajectory optimization.

**Grasp Switching:** Grasp switching is critical when a single grasp cannot maintain stability or reachability, such as opening a dishwasher with a handle positioned near the floor, making it inaccessible to the robot in one continuous grasp. To address this, we first sample an intermediate object state $\phi_{\text{mid}}$ between the start $\phi_0$ and goal $\phi_T$. We then solve for two sets of IK solutions: one using the initial grasp for $[\phi_0 \rightarrow \phi_{\text{mid}}]$ and the other using the final grasp for $[\phi_{\text{mid}} \rightarrow \phi_T]$. A short transition trajectory is planned between the two grasp configurations to enable collision-free detachment and reattachment. The three segments are concatenated into a continuous motion, yielding smooth trajectories with grasp switches executed only when necessary. This mechanism substantially expands task feasibility, as many articulated-object interactions (*e.g.*, dishwashers, drawers) cannot be completed under a single grasp.

### 4.4 DATA GENERATION

After trajectory optimization, we refine optimized trajectories to prone constraint-violated trajectories and synthesize realistic sensor observations for downstream tasks.

**Trajectory Post-Processing:** This stage verifies that each trajectory waypoint $x_{[t]}$ satisfies the required motion constraints. For fixed-base tasks, we compute the translational deviation $d = |p(x_{[t]}) - p(x_{\text{ref}})|$ and rotational deviation $\theta = \cos^{-1}\left(2 \langle r(x_{[t]}), r(x_{\text{ref}}) \rangle^2 - 1\right)$, where $p(\cdot)$ and $r(\cdot)$ denote the translational and rotational components of the AKR forward kinematics, and $x_{\text{ref}}$ is the reference configuration. For planar constraints, such as requiring motion constrained to the XY plane, we evaluate the vertical displacement $d_z = |p_z(x_{[t]}) - p_z(x_{\text{ref}})|$ and rool-pitch deviation $\theta_{\text{planar}} = |\psi(x_{[t]}) - \psi(x_{\text{ref}})|$, where $p_z(\cdot)$ is the z-axis translation and $\psi(\cdot)$ denotes roll and pitch. Trajectories violating any thresholded constraint are discarded. This process ensures all retained trajectories satisfy the specified kinematic constraints for stable, physically plausible execution.

**Multi-Modal Data Rendering:** We integrate both egocentric and fixed RGB-D cameras into each scene using NVIDIA Isaac Sim, configuring synchronized color and depth sensors on the robot and in the environment. At each trajectory waypoint, Isaac Sim renders high-fidelity RGB images and aligned depth maps, which are directly converted into point clouds in the simulation's coordinate

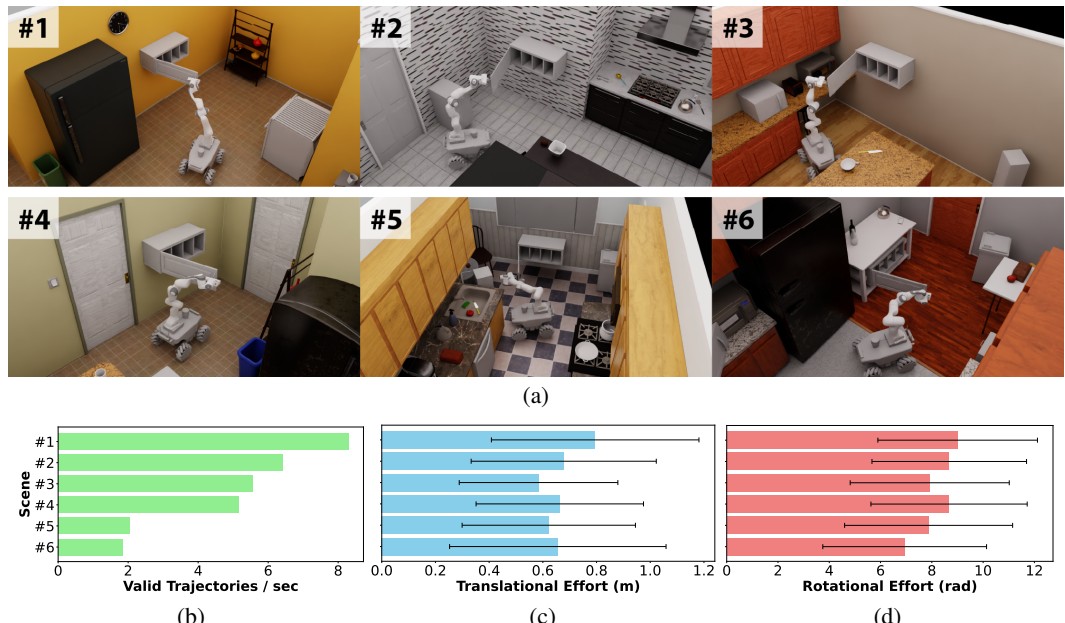

Figure 4: **Evaluation of trajectory generation performance across six representative household scenes.** (a) Visualizations of test scenes, with increasing confinement for realistic mobile manipulation. (b) Generation speed is measured as valid trajectories per second; simpler layouts result in higher throughput. (c) Average translational effort of the mobile base per trajectory, with error bars indicating standard deviation. (d) Average rotational effort of the manipulator, reflecting the compensatory motion required in constrained environments.

frame. Camera placements are fully customizable, and scenes can be re-rendered by replaying the generated trajectories. The resulting dataset supports a wide range of downstream tasks, including imitation learning (Fang et al., 2019; Hua et al., 2021), visual servoing (Sun et al., 2018; Janabi-Sharifi et al., 2011), and affordance detection (Chu et al., 2019; Do et al., 2018).

## 5 DATASET

### 5.1 TRAJECTORY GENERATION PERFORMANCE

To evaluate the effectiveness and generalizability of our trajectory generation framework, we conduct experiments across six representative kitchen scenes from the data environment. Each scene poses unique spatial constraints and increasing layout complexity (lower complexity means more collision-free IK could be found in that scene) (Fig. 4a). We deploy the Summit Franka mobile manipulator and execute a common articulated object manipulation task, opening a wall-mounted cabinet with an unwieldy door, in each environment.

We evaluate three key metrics:

- **Generation Speed:** Measured as valid trajectories (*i.e.*, trajectories that passed trajectory post processing) generated per second. The results are shown in Fig. 4b. Simpler layouts (*e.g.*, Scene #1 and #2) achieve higher data generation speed, while tightly constrained environments (*e.g.*, Scene #5 and #6) reduce generation speed due to limited spaces that constrain base movement.
- **Translational Effort:** Defined as the average distance traveled by the mobile base per trajectory. As shown in Fig. 4c, variations in base effort result in a diverse set of trajectories within the dataset.
- **Rotational Effort:** Measured by the cumulative angular motion of the arm. Similarly, Fig. 4d illustrates that variation in arm effort also contributes to the diversity of trajectories in the dataset.

These results highlight the adaptability of our planner to diverse spatial layouts and the ability to generate feasible whole-body trajectories in challenging, real-world scenarios.

## 5.2 Downstream Validation

To assess the utility of our dataset for training learning-based policies, we conducted downstream validation experiments under varying robot embodiments, scene diversity, and training set sizes. Tab. 2 reports the number of successful executions (out of 100 trials) for three representative policy classes: diffusion-based policy (DP), action-conditioned transformer (ACT), and diffusion policy with trajectory priors (DP3).

Table 2: **Downstream validation results.** Success counts (out of 100 trials) across robot embodiments and training set sizes.

| Embodiment | Training Size | ACT Successes | DP Successes | DP3 Successes |
|---|---|---|---|---|
| Fixed | 100 | 0 | 13 | 5 |
| Mobile | 100 | 0 | 0 | 6 |
| Fixed | 300 | 7 | 31 | 23 |
| Mobile | 300 | 0 | 0 | 18 |
| Fixed | 1,000 | 37 | 52 | 83 |
| Mobile | 1,000 | 2 | 13 | 55 |

**Effect of training data scale.** Across both mobile-based and fixed-based settings, increasing the number of training trajectories consistently improves success rates. For instance, in the mobile-based setting with a single scene, DP3 improves from 6 successes with 100 trajectories to 55 successes with 1,000 trajectories, while DP increases from 0 to 13 successes over the same range. This highlights the importance of large-scale data availability for generalizable mobile manipulation.

**Mobile vs. fixed embodiments.** Results reveal a persistent performance gap between mobile-based and fixed-based setups. With 1,000 trajectories in a single scene, DP3 achieves 83% success on the fixed-base but only 55% on the mobile-base. Similarly, DP and ACT obtain 52 and 37 successes respectively in the fixed-base, compared to only 13 and 2 in the mobile-base. This gap illustrates the amplified difficulty of coordinated whole-body planning when base motions are involved.

**Scene diversity.** To better evaluate the impact of environmental diversity, we test mobile-based DP3 across five distinct scenes using 5,000 training trajectories (1,000 per scene). The policy achieves only 22 successes in 100 trials, significantly lower than the 55 successes obtained with 1,000 trajectories in a single scene. This substantial performance drop highlights the challenges posed by scene diversity and underscores the need for both broader environmental coverage and larger data volumes to achieve robust generalization. Moreover, we conduct single-object generalization tests across 15 procedurally generated scenes, collecting 1,000 trajectories per scene, to systematically assess how scene count and data scale influence planning robustness.

Overall, these results demonstrate that current learning-based policies struggle to generalize across embodiments and diverse scenes without large-scale, physically valid data. By enabling efficient generation of such data, our dataset provides a critical resource for advancing robust mobile manipulation.

## 6 Limitations and Conclusion

AutoMoMa presents a scalable framework for generating large-scale, physically valid whole-body trajectories for coordinated mobile manipulation, producing over half a million examples across diverse scenes, objects, and robot embodiments. Despite its efficiency and extensibility, several limitations remain. First, the reliance on fixed layouts and known kinematic models restricts coverage of highly cluttered and outdoor scenarios. Second, the use of sphere-based collision approximations, while critical for GPU acceleration, can occasionally introduce geometric inaccuracies that lead to planning failures (see Appx. D). Finally, the current pipeline does not account for dynamic human–robot interaction or deformable object manipulation, which are important for real-world deployment. Looking forward, we plan to integrate learning-based components to further automate data generation and to develop community-driven tools that enable seamless extension with new robots, assets, and environments. These directions will broaden the applicability of AutoMoMa and strengthen its role as a foundation for advancing embodied AI.

## REPRODUCIBILITY STATEMENT

We have taken several measures to ensure the reproducibility of our work. A detailed description of the dataset generation pipeline, including scene layouts, object assets, robot embodiments, and motion planning algorithms, is provided in Sec. 4. To facilitate re-use and verification, we will release an **anonymous code repository** containing the GPU-accelerated planning scripts, trajectory post-processing modules, and rendering pipeline. The complete dataset of generated episodes, together with metadata (scene/task configuration files, robot models, and collision sphere parameters), will also be made available upon publication.

## LLM USAGE

We used large language models (LLMs), specifically OpenAI's ChatGPT, to assist in polishing the writing and improving the clarity of exposition. The LLM was employed exclusively for language refinement (e.g., grammar corrections, style adjustments, and conciseness), while all technical contributions, experiments, analyses, and claims were implemented and validated by the authors. No LLM-generated text was used without careful human verification, and the models did not contribute novel ideas, experimental results, or theoretical insights. Thus, the role of LLMs in this work was limited to aiding readability and presentation, similar to the function of a language editor.

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

## A  Augmented Kinematic Representation (AKR) Construction

### A.1  Collision-Sphere Fitting Procedure

As the manipulated object becomes part of the robot representation during planning via AKR, we approximate its collision geometry using a set of spheres. This procedure aligns seamlessly with cuRobo's sphere-based robot representation and enables GPU-accelerated parallel computation of collision checking.

Specifically, the collision-sphere fitting involves the following detailed steps:

1. **Mesh Preprocessing:** Since each link may contain multiple geometric components, these are first merged into a single link mesh.

2. **Mesh Scaling and Voxelization:** The merged link mesh is uniformly scaled down slightly to ensure conservative collision checking. This scaled mesh is then voxelized into discrete occupied volumetric regions, representing the shape of the object as an occupancy grid.

3. **Sphere Fitting:** For each occupied region identified through voxelization, an individual collision sphere is fitted. Each sphere is positioned at the centroid of the corresponding voxel region, with its diameter equal to the voxel edge length, thereby filling the voxel exactly.

4. **Spatial Alignment Adjustment:** Post voxelization, the resulting sphere cloud might experience translational offsets due to discretization. To maintain spatial consistency, we optionally realign the centroid of the fitted sphere cloud with the centroid of the original merged mesh, mitigating any significant translational drift introduced during voxelization.

The resulting compact, sphere-based collision representation ensures computationally efficient collision queries during trajectory optimization, crucial for maintaining interactive performance in our planning pipeline.

---

**Algorithm 1:** AKR Construction Procedure

---

1 **Function** `akr_construction` (*robot, object, init_state, scaling_factor, sphere_params, grasp_pose, grasp_link, sample_n*) **:**
2      `update_object_state`(object, init_state);
3      **foreach** *link ∈ object.links* **do**
4          merged_mesh ← `merge`(link.geometries);
5          scaled_mesh ← `scale`(merged_mesh, scaling_factor);
6          link.visual ← scaled_mesh;
7          link.collision ← `sphere_fit`(scaled_mesh, sphere_params);
8      **end**
9      **foreach** *joint ∈ object.joints* **do**
10          tf ← `get_tf`(object, joint.child, joint.parent);
11          joint.origin ← `update`(tf, scaling_factor);
12      **end**
13      scaled_object ← object;
14      fk_pose ← `fk`(scaled_object, grasp_link);
15      attached_origin ← grasp_pose$^{-1}$ · fk_pose;
16      inversed_object ← `inverse`(scaled_object, grasp_link);
17      akr ← `attach`(robot, inversed_object, attached_origin);
18      added_link_pairs ← `filter`(akr.link_pairs, robot.link_pairs);
19      sampled_cfg ← `sample_cfg`(akr, sample_n);
20      added_collision_pairs ← `check_collision`(added_link_pairs, sampled_cfg);
21      akr.collision_pairs ← `union`(robot.collision_pairs, added_collision_pairs);

---

### A.2  Detailed implementation for AKR inversion and assembly

To facilitate efficient integration of articulated objects into robotic manipulation pipelines, we present a structured workflow comprising the following steps: a) URDF preprocessing and kinematic inversion, b) collision spheres generation, c) object-to-robot attachment, and d) selective self-collision checking. The process begins by applying a uniform scaling factor to the object URDF to

ensure physical consistency. This step is particularly important when working with grasp datasets derived from point cloud or RGB-D data, where object models are often normalized to fit within a standardized bounding volume. Since each link may contain multiple geometric components, these are first merged into a single link mesh and scaled accordingly. As mesh scaling alters the spatial relationships defined in the original kinematic chain $\mathcal{K}_{\text{raw}}$, all joint origins are subsequently recalculated to preserve valid relative transformations. The resulting structure defines a new kinematic chain, denoted as $\mathcal{K}_{\text{scaled}}$.

The tip link $\ell_{\text{tip}}$, corresponding to the grasping point, is identified along with its parent joint. To enable attachment of the object as an extension of the robot, we invert the kinematic structure by reassigning $\ell_{\text{tip}}$ as the new base link. The joint hierarchy along the kinematic chain $\mathcal{K}_{\text{scaled}}$, from the original base $\ell_{\text{base}}$ to the tip, is reversed accordingly, while the rest of the tree structure is preserved. The resulting kinematic chain is denoted as $\mathcal{K}_{\text{inv}}$.

The transformation that defines the attachment between the robot and the object is computed based on the grasp pose and the object's forward kinematics(FK). Let $T_{\text{tip}}^{\text{base}}$ represent the pose of the selected tip link $\ell_{\text{tip}}$ frame relative to the object's original base link $\ell_{\text{base}}$ frame under the joint configuration $\boldsymbol{q}_{\text{init}}$ corresponding to the grasp pose.

$$T_{\text{tip}}^{\text{base}} = \text{FK}_{\mathcal{K}_{\text{scaled}}}(\boldsymbol{q}_{\text{init}}, \ell_{\text{tip}}) \tag{9}$$

The grasp pose is denoted as $T_{\text{tcp}}^{\text{base}}$, which typically specifies the pose of the robot's TCP frame with respect to the object's base link frame. The final attachment transformation is computed as:

$$T_{\text{tip}}^{\text{tcp}} = \left(T_{\text{tcp}}^{\text{base}}\right)^{-1} \cdot T_{\text{tip}}^{\text{base}} \tag{10}$$

The transformation is applied as a fixed joint between the robot's TCP and the object's new base link (formerly the selected tip link $\ell_{\text{tip}}$), resulting in a unified kinematic model that integrates the robot and the object into a single tree structure.

Finally, we identify the additional self-collision link pairs introduced by the attached object, avoiding a full recomputation of the entire self-collision matrix. This selective check reduces computational overhead while ensuring sufficient coverage for motion planning and safety checks.

## B  IMPACT OF GRASP-SWITCHING (VS. FIXED-GRASP)

To evaluate the effect of grasp-switching on manipulation performance, we compare two trajectory sets: one using a fixed grasp throughout the task and another allowing grasp-switching when beneficial. Figure 5 presents example opening angles achieved by each method.

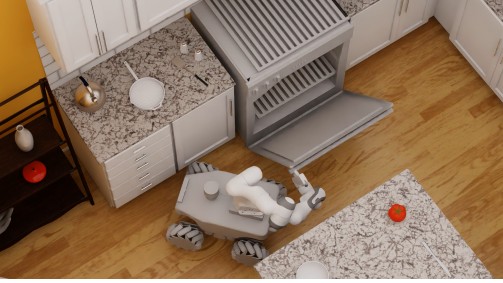 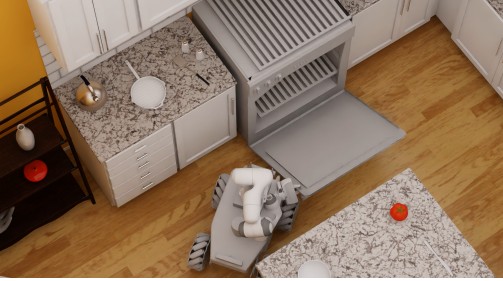

(a) Fixed-grasp trajectory.        (b) Grasp-switching trajectory.

Figure 5: Comparison of object opening angles under fixed-grasp and grasp-switching strategies.

In these examples, grasp-switching enables the robot to change the grasp pose, which increases the achievable opening angle by avoiding link collisions. Quantitatively, we observed that trajectories with grasp-switching attained larger maximum opening angles compared to fixed-grasp baselines, demonstrating the importance of grasp-switching in constrained manipulation tasks.

## C  GENERATION OF INTERACTIVE SCENES

AutoMoMa leverages two complementary sources of interactive household environments: manually curated scenes and procedurally generated layouts.

The first source consists of 30 high-fidelity scenes derived from AI2-THOR (Kolve et al., 2017) (shown in Fig. 6a). In each, we manually replace static objects such as microwave ovens, dishwashers, and cabinets with functionally equivalent articulated counterparts from the SAPIEN dataset (Xiang et al., 2020). These replacements are carefully positioned to respect semantic context and physical plausibility, yielding semantically coherent scenes ideal for targeted evaluation.

To enable a large-scale dataset for the downstream task, we generate an additional 300 diverse scenes using a custom Infinigen-based pipeline (Raistrick et al., 2024). We convert the articulated object models into static placeholder assets and import them into Infinigen. Our generator supports controllable parameters—including object selection, placement optimization, and layout sparsity—to guide procedural generation toward manipulation-friendly configurations. The global layout is optimized while ensuring each selected articulated object remains embedded in the final scene. The resulting layouts are exported in USD format for compatibility with GPU-accelerated planning in cuRobo (Sundaralingam et al., 2023), after which placeholders are replaced with the original articulated objects to restore full kinematic fidelity, shown in Fig. 6b.

Together, these two complementary scene sources—30 curated and 300 procedural—provide a rich and diverse foundation for training and evaluating robust mobile manipulation policies across a wide range of household contexts.

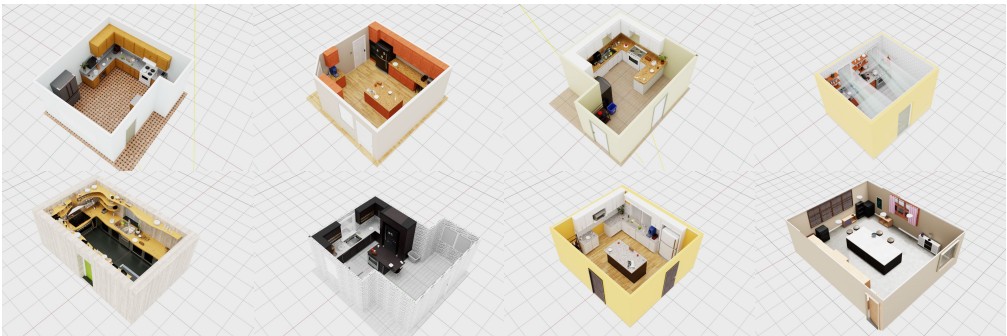

(a) AI2-THOR(iTHOR) scene with asset replacement.

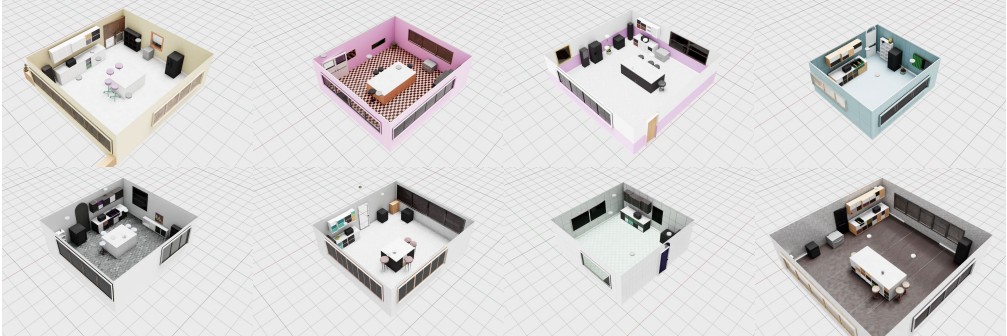

(b) Procedurally generated scene from Infinigen.

Figure 6: Two approaches for building interactive environments in AutoMoMa: replacing using SAPIEN assets in AI2-THOR(iTHOR) and generating layouts with Infinigen.

## D  REPRESENTATIVE SUCCESS AND FAILURE CASES

We provide visual examples of both successful and unsuccessful trajectories generated by our pipeline, highlighting common issues encountered during planning.

**Failure due to Collision:**   Figure 7 illustrates a trajectory that resulted in a collision. Such collisions primarily occur because the robot and manipulated object representations are simplified as sphere-based models for computational efficiency. This spheroidization can occasionally lead to inaccuracies where the simplified geometry fails to precisely capture the original shape, resulting in unintended collisions during planning.

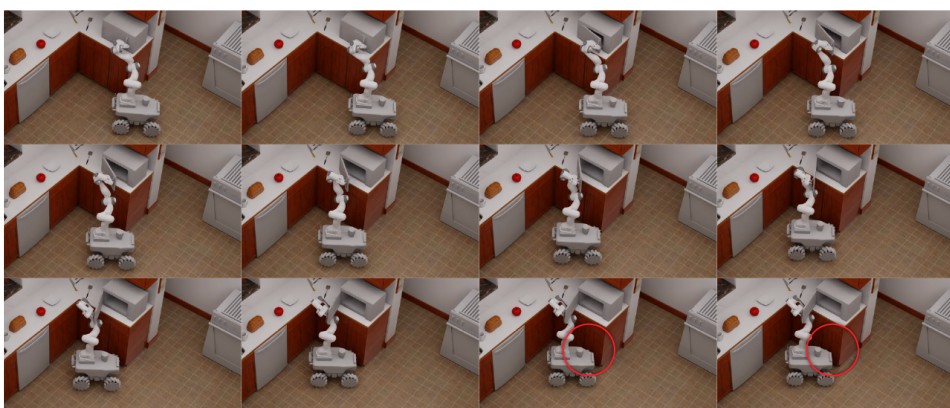

Figure 7: Example of trajectory failure due to collision from simplified robot model.

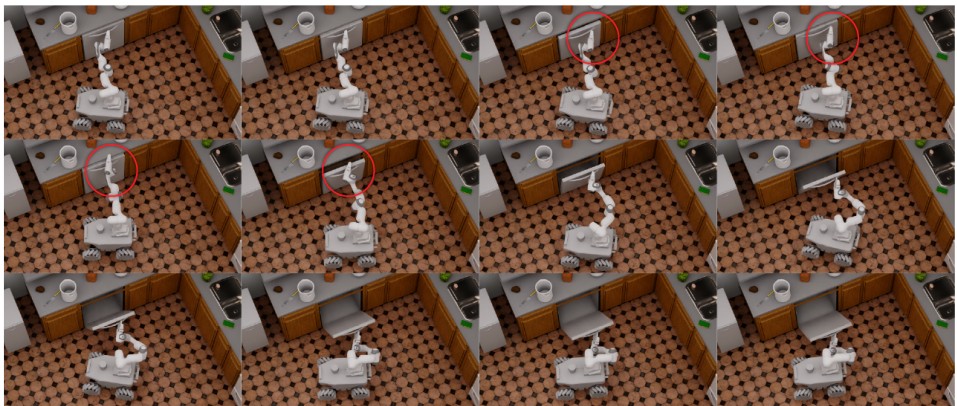

Figure 8: Example of trajectory failure due to fixed-base constraint violation.

**Failure due to Constraint Violation:**   Figure 8 depicts a trajectory that violates constraints, specifically a fixed-base constraint violation. These constraint violations typically arise when the planned trajectory erroneously involves movements inconsistent with object fix-base constraints defined in the planning task. Ensuring strict adherence to constraints remains challenging, particularly in complex manipulation scenarios.

**Successful Trajectory:**   Conversely, Figure 9 showcases an example of a successful trajectory, demonstrating how effective trajectory generation appropriately respects both collision constraints and the specified motion constraints. This example validates the pipeline's capacity to generate feasible and physically realistic robot motions, emphasizing the pipeline's utility in diverse manipulation tasks.

## E   FAILURE CASES OF LEARNED POLICIES

Despite achieving promising performance, policies trained on our dataset exhibit common failure modes when deployed over long horizons. The most prevalent issue arises from the accumulation of small prediction and execution errors, which progressively amplify into significant end-effector drift and ultimately cause task failure. Figure 10 illustrates two representative examples.

## F   REAL-WORLD VALIDATION

We validate our planning pipeline on a physical UR5-Ridgeback system, which comprises two UR5 manipulators mounted on a Clearpath Ridgeback mobile base. Two representative tasks were tested: opening a drawer and opening a cabinet door. In both tasks, the robot executed the planned trajectories smoothly, accurately reproducing the motion patterns generated in simulation without collisions or constraint violations.

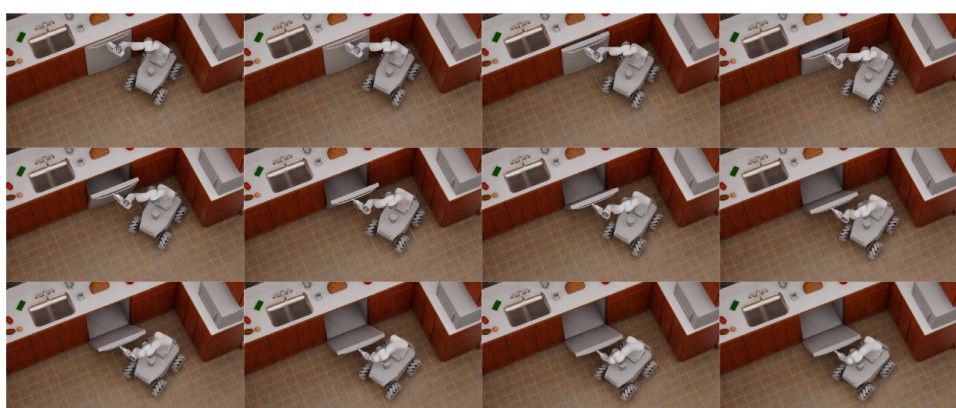

Figure 9: Example of a successful trajectory.

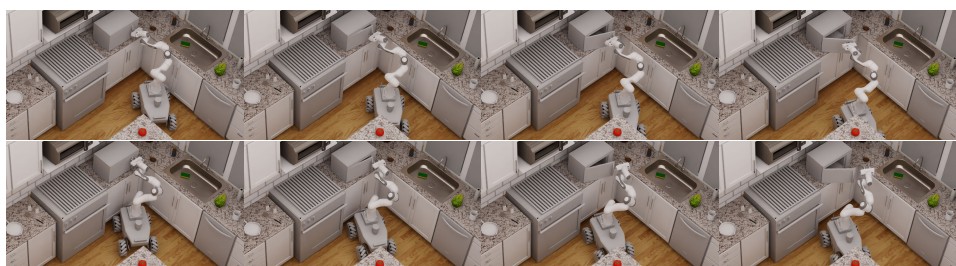

Figure 10: **Failure cases of mobile-base policies.** Small prediction and control errors accumulate over time, leading to drift, collisions, and eventual task failure.

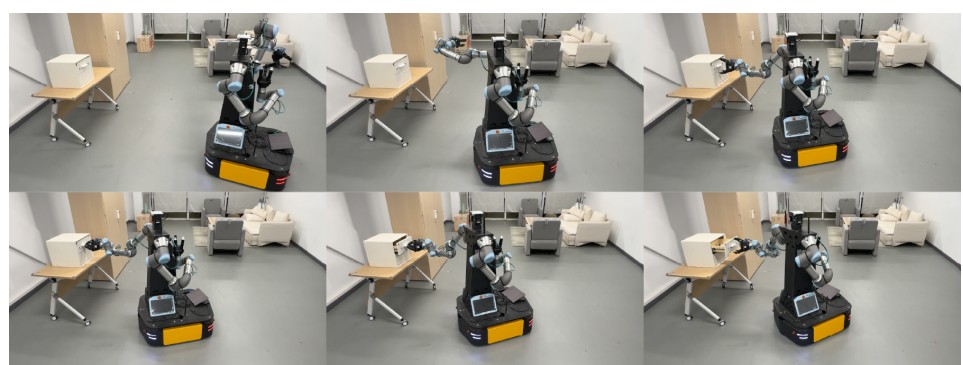

Figure 11: UR5-Ridgeback executing a planned drawer-opening trajectory.

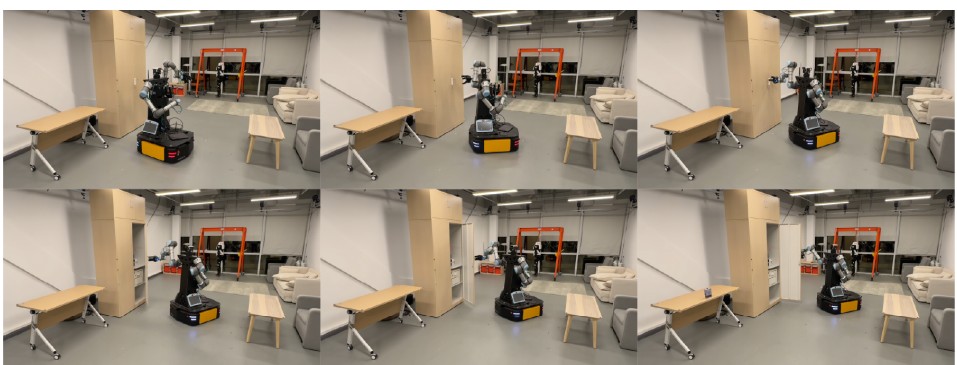

Figure 12: UR5-Ridgeback opening a cabinet door using the planned motion.

