# OpenReview forum: "AutoMoMa: Scalable Coordinated Mobile Manipulation Trajectory Generation"
_ICLR.cc/2026/Conference — ICLR 2026 Conference Withdrawn Submission_

### Official Review · Reviewer_9F6u · 2025-10-31

**Soundness:** 3
**Presentation:** 3
**Contribution:** 3
**Rating:** 4
**Confidence:** 4

**Summary:**

This paper introduces a GPU accelerated planning system that generates simulation trajectories far faster than prior methods in diverse scenes and tasks. It introduces a method to construct a automatic kinematic representation (AKR) of the mobile manipulation base, robot arm, and target object. An optimization problem is then solved to go from an initial AKR state to a goal AKR state, generating a mobile manipulation trajectory.

**Strengths:**

- This work introduces a novel method to generate via GPU accelerated motion planning trajectories for mobile manipulation. To my knowledge there currently aren't any alternatives that run at reasonable speeds and past work has typically relied on hacky solutions or learning methods.
- The trajectory generation speed is far faster than prior work (80x) which is impressive and can be easily understood why it is achieved here.

**Weaknesses:**

- Some more details regarding the drop in performance when using mobile base vs fixed base would be helpful. I'm of the understanding that some tasks such as opening articulated doors would be easier if the mobile base can be used. At minimum the task should be solved faster if the policy does succeed. Some more information would be appreciated since if the mobile base is not useful then this goal of mobile manipulation is a bit diluted.
- A small concern is the limitation to modeling collisions as spheres only. For some more concave shaped objects this would not be possible and would limit the possibilities of this work.
- While the paper wants to claim scalability and the paper does present how it is scalable, the experiments only use up to 1000 demonstration trajectories whereas past work uses far more. I only see that in table 1 it is mentioned there are 500,000 trajectories generated. However since this is simulation you could generate as much as you want fairly quickly. Thus it's unclear whether more data is helpful and whether the scalability is working.

**Questions:**

- The supplemental videos only show examples where the robot is already grasping a target object e.g. a handle. Are there details for how data to get to the grasping state is generated?

---

### Official Review · Reviewer_JSJj · 2025-10-31

**Soundness:** 2
**Presentation:** 2
**Contribution:** 2
**Rating:** 2
**Confidence:** 2

**Summary:**

The paper proposes a data generation approach for mobile manipulation based on motion planning in simulation.

**Strengths:**

Scalable data generation approaches for robotics are an important research area.

**Weaknesses:**

- The empirical evaluation is limited overall. It is a bit hard to interpret and understand the results of the proposed approach. It is also not clear if the primary contribution is a dataset or a method for generating a dataset. If former, the approach should be compared to alternate data generation approaches. If the latter, to alternate methods for generating motion trajectories.
- The writing is not very clear and specific. For example, it is very hard to to tell much about the approach and contributions from the abstract. The introduction enumerates contributions but does not put them in context with respect to baselines or prior approaches which makes it hard to understand. I would encourage the authors to be specific and describe the work in simple concrete terms.
- It would be good to comment on the generalization of the approach to different robot morphologies. Majority of the experiments use a wheeled robot base with an arm. It would be good to comment on how this approach could be applied to different morphologies like dogs or humanoids.

**Questions:**

Please see the weaknesses above.

---

### Official Review · Reviewer_7WZc · 2025-11-01

**Soundness:** 3
**Presentation:** 4
**Contribution:** 3
**Rating:** 6
**Confidence:** 3

**Summary:**

The paper proposes AutoMoMa, a GPU-accelerated mobile manipulation data generation pipeline. Thanks to the high-throughput pipeline, authors are able to provide over 500k trajectories across diverse scenes, embodiments, and tasks. The trajectories are used to train imitation learning policies, and the realism of the trajectories are validated on a real robot setup.

**Strengths:**

- High throughput allows for large-scale data generation compared to prior works
- Tasks are non-trivial for mobile manipulation, including cluttered scenes with many possible points of collision, and tasks which require grasp-switching
- Generated demonstrations included coordinated whole-body control, a relevant research area in the community
- Realism of generated trajectories are validated on a real robot

**Weaknesses:**

- The experiments on learned policies are somewhat limited, i.e. the trained policies do not fully leverage the large dataset size, or demonstrate improvements via mobile manipulation
- The real robot evaluations do not evaluate trained policies, only generated trajectories; hence, the generated data itself may be insufficient for sim2real transfer of learned agents

**Questions:**

1. While the dataset provides 500k trajectories, the policies are only trained with up to 1k trajectories (or 5k for cross-scene policies). Do the authors have results leveraging larger dataset sizes (e.g. does performance plateau after ~1k trajectories per scene)?
2. Per the evaluations, the stationary manipulation policies outperform mobile manipulation policies. Did the authors discover a reason for this? It could indicate the tasks are too constrained, or the generated trajectories have insufficient coverage over the state-action space.

---

### Official Review · Reviewer_8KNx · 2025-11-01

**Soundness:** 3
**Presentation:** 2
**Contribution:** 2
**Rating:** 4
**Confidence:** 5

**Summary:**

This paper proposes AutoMoMa, a system for generating coordinated mobile manipulation trajectories at scale. The method builds on the Augmented Kinematic Representation (AKR) framework and further accelerates trajectory generation using GPU-based motion planning. In the reported benchmark scenarios, AutoMoMa can generate approximately 5,000 trajectories per GPU hour.

**Strengths:**

- The paper is clearly written, well-organized, and easy to follow.
 - The problem addressed, scalable data generation for mobile manipulation, is important and timely for the field.

**Weaknesses:**

- The evaluation focuses on relatively simple single-stage tasks. For example, in the supplementary videos, the robot often starts with the gripper already placed on the target handle. This setup reduces the complexity of planning and does not reflect realistic multi-stage, long-horizon household tasks where navigation, approach, and grasp phases must be coordinated.
 - Data diversity is not fully discussed. While the method is scalable, the diversity of the generated data is not clearly analyzed. It would be valuable to explicitly quantify or visualize diversity in scene layouts and object configurations, base pose configurations, end-effector trajectories and manipulation strategies.
 - Important prior work is not cited or discussed in sufficient depth. For example, MoMaGen (for long-horizon bimanual mobile manipulation data generation) and MimicGen (for generating demonstration datasets) are closely related. A comparison would help clarify the unique contributions and limitations of AutoMoMa.
     - https://momagen.github.io/
     - https://mimicgen.github.io/

**Questions:**

- How are the initial robot states and object configurations randomized across episodes?
 - Can you provide visualizations or statistical analyses to demonstrate the diversity of the generated trajectories and scenes?
 - Can the system handle long-horizon tasks where the robot must first navigate to the object before manipulation (e.g., moving from across the room to the oven, grasping the handle, then opening it)?
 - How does AutoMoMa compare to MoMaGen in terms of task scope, data quality, scalability, and diversity?
 - What sensory observations are used as policy input? Are the experiments based on ego-centric views, third-person views, or both?
 - Do you have results or discussions regarding the real-world transfer of policies trained on AutoMoMa-generated data?

---

### Note · Authors · 2025-11-12

**Comment:**

We have decided to withdraw this submission from consideration. We would like to express our sincere appreciation to the reviewers for their thorough and constructive feedback. The comments and suggestions have provided valuable insights that will greatly help us strengthen and refine our work in future revisions.

**Withdrawal Confirmation:**

I have read and agree with the venue's withdrawal policy on behalf of myself and my co-authors.

---

### Note · Authors · 2025-11-12

**Comment:**

We have decided to withdraw this submission from consideration. We would like to express our sincere appreciation to the reviewers for their thorough and constructive feedback. The comments and suggestions have provided valuable insights that will greatly help us strengthen and refine our work in future revisions.

**Withdrawal Confirmation:**

I have read and agree with the venue's withdrawal policy on behalf of myself and my co-authors.